# Whole-Body Photobiomodulation Therapy Propels the Fibromyalgia Patient into the Recomposition Phase: A Reflexive Thematic Analysis

**DOI:** 10.3390/biomedicines12051116

**Published:** 2024-05-17

**Authors:** Bethany C. Fitzmaurice, Rebecca L. Grenfell, Nicola R. Heneghan, Asius T. A. Rayen, Andrew A. Soundy

**Affiliations:** 1Department of Pain Management, Sandwell and West Birmingham NHS Trust, Birmingham B18 7QH, UK; arasu.rayen@nhs.net; 2School of Sport, Exercise and Rehabilitation Sciences, University of Birmingham, Birmingham B15 2TT, UK; n.heneghan@bham.ac.uk (N.R.H.); a.a.soundy@bham.ac.uk (A.A.S.); 3Clinical Research Facility, Sandwell and West Birmingham NHS Trust, Birmingham B18 7QH, UK; rebecca.grenfell1@nhs.net

**Keywords:** fibromyalgia, photobiomodulation, experience, recomposition

## Abstract

Background: Recent evidence has identified great promise for the novel whole-body photobiomodulation therapy (PBMT) for individuals with fibromyalgia (FM). However, currently no evidence has documented the experiences of participants. The objective of this study was to qualitatively assess treatment experience and response in a group of participants with FM undergoing a course of whole-body PBMT. Methods: An interpretive hermeneutic phenomenological study situated within the worldview of pragmatism was undertaken. A convenience sample of individuals with FM were included if they had undertaken a novel 6-week trial of PBMT. Individuals undertook semi-structured interviews exploring treatment experience and multidimensional treatment responses during Week 3 and Week 6. Results: Sixteen trial participants (47.3 ± 10.9 years) took part in this study. The analysis produced three overarching themes that were previously identified from a baseline study (namely, ‘Body Structure & Function’, ‘Activities & Participation’, and ‘Environment’) with an additional five sub-themes that highlighted the intervention experience. Subsequently, four important processes were observed and identified: increased motivation; feeling proud; improved confidence; feeling like ‘old self’. This ultimately culminated in the identification of a positive spiral, which we have termed ‘recomposition’. Conclusions: We believe our study is the first in the field of chronic pain management to utilise qualitative methodology to directly assess the acceptability and efficacy of a specific medical intervention in a clinical trial, and the first study to qualitatively assess whole-body PBMT experience. The findings are compelling and warrant further work to support the introduction of this device into the National Health Service (NHS).

## 1. Introduction

Fibromyalgia (FM) syndrome is characterised by persistent and widespread pain associated with intrusive fatigue, sleep disturbance, impaired physical and cognitive function, and psychological distress [1]. In practice, this translates into a non-exhaustive list of additional and often unrealised symptoms such as hypersensitivities to sound, light, temperature, disproportionate tenderness, headaches, urinary symptoms, bowel symptoms, and dysmenorrhoea [1], to name a few. In fact, there are numerous online resources and supporting charities for patients, unveiling a list of over 250 symptoms [2]. Regardless of whether dubbed as primary or secondary FM, it has been shown that symptom burden is equivocal [3]. Its aetiology is multifactorial, with some of the proposed mechanisms including peripheral and central nervous system sensitisation, genetics, distress in the form of a major physical or psychological trauma [1], and mitochondrial dysfunction [4,5]. Understandably, the complexities of presentation are overwhelming, not just for the patient but indeed for the clinician [6]. ‘Ineffective treatments so far’ are commonplace, and as such, this factor has made it into the recent Royal College of Physicians’ national guideline of ‘fibromyalgia syndrome alert factors’, when prompting clinicians to consider FM as the diagnosis [1].

To date, a multitude of qualitative data exists within the field of chronic pain management that describes experiential elements of an individual’s pain condition [7]. With regards to FM, this specifically includes lived experiences [8,9,10], experience of FM flares [11], experience of being diagnosed with FM [12], and generic experience of previously tried FM treatments [8,9]. In this regard, understanding the FM experiences i relatively well understood. However, further recognition and understanding of treatment experiences are needed, particularly towards the development of new services [13]. It is documented that marginalised populations are not meaningfully engaged in health service planning [14], with the FM population being one such example [15].

Limited evidence has examined the experience of new treatments. One evidence synthesis describes the complexities of physiotherapy engagement [16], while a large qualitative systematic review of complementary and alternative therapies for FM summarises respective randomised controlled trials [17]. Only two qualitative studies assessing medical treatment acceptability and effectiveness across chronic pain management as a whole were identified: exploring patient perspectives and experiences of opioid medicines [18] and interventional pain management [19], respectively. The methodology of the former utilised qualitative evidence synthesis [18]. Therefore, the only qualitative study to directly examine a response to a pain intervention, being a small sample of participants, describes the overall experience of the healthcare encounter only [19], not the specific treatment experience nor response. It does, however, provide valuable insight into mechanisms of change and highlights the value of qualitative experiences.

Honing back in on the FM-specific response to an intervention, a large (*n* = 728) qualitative synthesis optimistically entitled ‘FM pain management effectiveness from the patient perspective’ [20] describes an array of important, but generic, problematic issues: current management options are a continued source of frustration where patients feel stigmatised and overmedicalised, and there continues to be unmet needs in physical, psychological, and pharmacological management. Indeed, the only FM study identified that directly and qualitatively assesses a response to an intervention describes the impact of peer social support groups [21]. Hence, very little qualitative research has been conducted in order to directly assess the response to a medical treatment in the field of pain management, and certainly none with respect to FM.

Quantitative data to date [22,23], albeit with a small sample size, shows huge potential for whole-body photobiomodulation therapy (PBMT) to not only aid in managing all FM symptom domains but, in some cases, to render the individuals no longer meeting the FM diagnostic score when reassessed post-intervention. However, quantitative data has outcome deficiencies, which are also recognised within best practice documents [24,25].

There is limited understanding around the experiences of treatment with PBMT, with limited knowledge around changes from pre- to post-intervention. To date, only one past piece of qualitative research (*n* = 11) describes patient experience with PBMT; however, this was in an alternative condition (head and neck chronic lymphoedema) [26]. Furthermore, National Institute for Health and Care Excellence (NICE) 2021 research recommendations on chronic primary pain [27] (an umbrella term encompassing fibromyalgia syndrome [28]) and PBMT identify prior outcome deficiencies. Our data addresses this NICE-stipulated research gap.

Past qualitative research has revealed the importance of qualitative data in establishing how, when, and why patients experience changes in their condition [26]. This type of data can meet the needs of the current topic [25] and is well-placed to consider if PBMT can act to improve the decomposition process [29].

Research that is able to provide insight into potential changes and processes is needed; qualitative research is ideally placed to achieve this. Research so far has shown that pain’s subjective and multidimensional nature is best evaluated utilising qualitative methodology in order to capture the richness of the patient’s experience [25], not only in relaying lived experience but also to examine treatment responses. The objective of this study was to qualitatively assess treatment experience and response in a group of participants with FM undergoing a course of whole-body PBMT, with the aim of supplementing the results of quantitative data from the same study population [22] in order to support a future definitive trial.

## 2. Materials and Methods 

The Standards for Reporting Qualitative Research (SRQR) [30] were used to report this section. Additionally, in order that the intervention can be replicated when building on future research, the TIDieR (Template for Intervention Description and Replication) [31] checklist was utilised.

### 2.1. Qualitative Approach and Research Paradigm

A qualitative approach was utilised to best explore participants’ experiences, perceptions, and opinions [32] regarding their response to whole-body PBMT. Pain experience is inherently subjective and “fundamentally unobservable” and thus deserves to be assessed comprehensively and multimodally [25].

### 2.2. Researcher Characteristics and Reflexivity

Researcher 1 (B.C.F) has a background in anaesthetics and chronic pain management. B.C.F led the data collection phase. Researcher 2 (A.A.S) is a qualitative research methodologist who had no direct contact with trial participants. Researcher 3 (R.L.G) is a clinical research practitioner who assisted with data collection. Researcher 4 (N.R.H) is a senior researcher in musculoskeletal rehabilitation sciences with a clinical background as a practicing physiotherapist. Researcher 5 (A.T.A.R) has a background in anaesthetics and chronic pain, having practiced as a consultant in chronic pain management for over twenty years to date. We decided at the outset that there would be more than one data reviewer in order to maximally utilise our broad range of strengths.

### 2.3. Context

All audio-recorded, semi-structured interviews were one-to-one and in person. They were conducted at Sandwell General Hospital’s Clinical Research Facility, West Bromwich, UK. The interviews took place over two phases, the first being mid-treatment after treatment visit 9; the second phase was undertaken at the final visit. All interviews were undertaken before the individual treatment session took place.

### 2.4. Sampling Strategy

From January to June 2022, a non-probability convenience sample was recruited from the Department of Pain Management at Sandwell and West Birmingham NHS Trust, Birmingham, UK. For the purposes of being eligible to receive whole-body PBMT, prospective participants were required to satisfy all eligibility criteria demonstrated in Table 1. The data collection process was piloted with a sample of five participants (A–E) during the first round of eight participants. Following Trial Steering Committee meetings, it was decided to continue with data collection while new themes continued to emerge with respect to the primary interviews. Interviews were completed with all remaining participants to ensure data saturation was reached.

### 2.5. Ethical Issues Pertaining to Human Subjects

Ethics approval was granted by the Health Research Authority (HRA) and Health and Care Research Wales (HCRW) (278452) and the Leicester Central Research and Ethics Committee (21/EM/0231) (ClinicalTrials.gov trial registration number NCT05069363; 06/10/2021). All participants gave written informed consent and were free to withdraw from study participation at any point. At each interview visit, it was re-iterated that the interview was wholly optional, and that the participant could stop the interview and recording at any point.

### 2.6. Data Collection Methods

Participants who opted to be interviewed during the consent process underwent semi-structured audio-recorded interviews at three stages of the trial; ‘pre-intervention’ at their first visit [29], ‘during intervention’ at Treatment Visit 9, and ‘post-intervention’ at their final visit. Interviews took place from 31st January 2022 through to 29th June 2022. The current study focuses on the trial experiences mid- and post-intervention: namely, 18 sessions of whole-body PBMT (at three and six weeks, respectively). The interview topic guide was developed by experienced qualitative researchers based on research objectives (see Appendix A)—the topic guide needed to be tailored in this instance due to the niche area being assessed and hence the lack of validated interview topic guides in this area. The ‘during intervention’ interview section was relatively short in comparison with pre- and post-intervention schedules, comprising questions firstly pertaining to the intervention and secondly with regard to trial design, conduct, and processes (four main questions). Subsections of the ‘post-intervention’ interview expanded on current symptoms and quality of life, medication usage, and then about the trial and the whole-body PBMT device (nine main questions). The interview schedule remained consistent for the duration of the trial. No adaptations to the questions were made. Demographic data was collected both verbally and via clinical records with the participant’s consent.

### 2.7. Data Collection Instruments and Technologies

Audio recordings were made via Microsoft Teams Version 10.4.7 for purposes of transcription. The primary researcher (B.C.F.) collected 87.1%, of the data for the mid- and post-treatment dataset, with the remainder being collected by three trial investigators when the primary investigator was not available. Mid-treatment interviews lasted an average of 21 min, ranging between 9 and 43 min. Final visit interviews were an average of 24 min; range 6–46 min. The combined average length was 22 min.

### 2.8. Data Processing

Interview data was transcribed by authors B.C.F and R.L.G. The authors primarily adopted a reflexive thematic analytical approach [33,34], supplemented by guidance on quality [35]**,** and a pragmatist approach [36], which was felt to be suited to both the data set and target audience. The raw data and transcripts were stored on a secure, password-encrypted NHS computer. For the purpose of the write-up, the participant’s trial codes were further anonymised with an alternate coding system.

### 2.9. Data Analysis

For purposes of analysis, mid- and post-treatment interviews were not differentiated as they were separated by only a small timeframe. As with the reflexive thematic analysis for baseline interviews [29], we describe our six-phase analytical process [34] in Appendix A, namely “Familiarisation with the data”, “Generating initial codes”, “Generating themes”, “Reviewing potential themes”, “Defining and naming themes”, and “Producing the report”.

### 2.10. Techniques to Enhance Trustworthiness 

Table 2 gives a detailed description of the trial intervention so that future trials can be easily replicated, as stipulated by the British Medical Journal [31]. The trial was monitored and audited by Sandwell and West Birmingham NHS Trust’s Research Management and Governance Facilitator pre-, mid-, and post-trial. An audit trail was documented throughout the analysis, with each amendment being saved as its own document so that all processes are transparent. 

## 3. Results

### 3.1. Description of Participants

The present study reflects qualitative data from 16 trial participants, out of a potential 19. Three participants were not interviewed due to the simultaneous and ongoing evaluation of pilot data. Table 3 depicts the participants’ demographic data. A further breakdown of the data specific to each participant can be found in our baseline results [29].

The following results are represented according to the World Health Organization’s International Classification of Functioning, Disability, and Health (ICF) Brief Core Set—“body structure and function”, “activities and participation”, and “environment” [37]. Notably, not all participant quotes are included, but rather the focus is on important themes. Most sections conclude with a handful of additional important and indictive quotes. A further breakdown of all relevant quotes and from whom can be found in Appendix A.

### 3.2. Themes

Each theme encompasses both mid- and post-intervention interviews. All except one participant that were interviewed at baseline received two further interviews, with one participant missing the mid-treatment interview. One participant was excluded from the dataset because despite completing all treatments, he was not able to spare time for post-intervention outcome measures and was subsequently lost to follow up. Where appropriate, comments and themes are compared to the baseline [29] for further contextualisation. Results are presented in order of commonality, making it theoretically easier for the reader to deduce the domains with the most significant potential impact. Our three prior overarching themes, according to the ICF Brief Core Set, are consistent with this analysis (Figure 1). A separate five sub-themes arose in relation to intervention experience.

#### 3.2.1. Body Structure and Function

1: Improvement in pain. All but one participant describes a relief from pain, described as a reduction of 70–100% (C;H;J;K;M) or 5 out of 10 instead of 9 out of 10 (I), and as low as 2–3 out of 10 for the first time in years (N), nothing like it was (K), and being bemused by a lack of pain (J;L;M). One participant (N) describes *“I didn’t have pain in places I ain’t had for ages…for the last 3 or 4 years, put it like that”.* This included pain being generally less intense (D;E;G;J;L;M;N;O;P), less waking up at night due to pain (C;J), reduced frequency and severity of headaches (B;H), pain in legs alleviated (C;O)—*“doesn’t feel like two lead weights”*, no longer getting cramps and pulsation pains (C), less tender at certain points (D), including shoulders, neck, spine, knees, feet, elbows, hands, and fingers (G;J;L;M;N;O;P). Pain can be present daily but tolerated better (A) and manageable (B;C), and consequently, patients expressed an ability to engage in meaningful activities (A;B;C;I), including travelling long distances to visit relatives for the first time in years (I). Reduced pain had significantly positive knock-on effects on factors affecting quality of life, with one participant (D), who has always had to wear softer and looser sports bras, now being *“able to put a real bra on”.* Pain is no longer so intrusive in the way it controls aspects of life, and participants are recognising they are *“not complaining about it as much”* (E). Skin sensitivity pain has completely evaporated to *“zero”* in a participant who previously described a “*chisel digging down in flesh”* and *“like a hot piece of iron on skin”* (H).

*“The pain’s there but it’s not intense so I’ve been able to do things that I haven’t done in a long time”* *(Participant A)*

*“loads better…I can’t remember the last time my pain was this low”* *(Participant E)*

*The other aches and pains in my shoulders, knees and my feet—that seems to melt away quite quickly”* *(Participant G) *

*“It normally burned and could feel crawling sensations, and trigger points would be buzzing…this week I can say it’s zero, I don’t have it”* *(Participant H)*

*“Walk…doing a lot of art…which has been easier with less pain”* *(Participant I) *

*“A lot of the pain has gone from my shoulders and neck completely, I just think—‘why isn’t that hurting me?’”* *(Participant J)*

*“my pain—I went in there [PBMT device] and I thought ‘bloody hell, this is good’”* *(Participant L)*

*“it was just excruciating, but like I say since this [PBMT] I can’t believe there’s just no pain there you know”* *(Participant M)*

2: Reduced lethargy and fatigue. Similarly, all but one participant describes increased energy levels, feeling more refreshed, and/or a reduction in fatigue/exhaustion/feeling ‘drained’. One participant (H) describes no longer suffering any fatigue whatsoever. Another (I) objectified to her fatigue score out of 10 being at an all-time low (2, compared with the usual 9). Reduced fatigue was described as both physical and mental. Participants (D;G;O) describe feeling less heavy, achy, lethargic, and ‘fed up’ when waking up, but rather more alert and awake. This was seconded by a further participant (E), who described *“brain is more awake”*. In some cases (E;J;O), increased energy translates into the omission of daytime naps and continuing daily activities despite being busy the night before, compared with previously requiring a ‘rest day’. Participants attribute this fatigue reduction to being able to do more—*“on the Friday, I done the house from top to bottom, so work that one out!”.* This particular quote was midway through the trial intervention, in a participant that usually struggles to muster up the energy for basic personal hygiene. Participants (B;C) are beginning to demonstrate an insight into symptom interlinkage: improved energy consequently improving other aspects of their life, including ‘their whole day’, being able to increase work productivity, as well as feeling more motivated about it—*“the energy has given me the motivation to do the stuff”.* There is a recognition that less tiredness has meant fewer aches and pains (C) and that increased energy has translated into an increase in positive habits, such as now being able to both manage and want to sit in the garden for an hour (F).

*“I’ve got that bit more burst of energy”* *(Participant G)*

*“I don’t seem to have so much fatigue in my legs, they’re not so tired, same with my arms—they’re not so fatigue-y”* *(Participant M)*

3: Improved sleep. Participants feel more relaxed prior to and during sleeping (A;D;G;J) even in participants that describe sleep as their most intrusive FM symptom (P), identified getting to sleep much easier (A;B;C;E;G;H;I;J;L;O;P) without having to pad themselves out with multiple pillows (J) or overthinking (D;J;O), feel ready for bed when they go (B;D;P)—*“physically, I can feel my body’s able to rest, whereas before it wasn’t”*, sleep is better quality and more restful (B;D;E;G;H;I;J;P), and managing to sleep longer (A;B;C;D;E;G;H;I;M;P) and in some instances all the way through until morning (B;E;J;K), less broken sleep (C;E;I;J;M;P), specifically due to less pain (C;E;J;K), less spasms and restlessness (D), easier to return to sleep if woken (C;O). One participant (C) attributed improved sleep to attenuated leg pains. The same participant continues to wake in the night, albeit much less frequently; prior to trial, she was waking every hour and sometimes could not get back to sleep whatsoever. Now, he can go for five hours, and *“9 times out of 10 now I can fall back to sleep”.* Participants feel more refreshed in the mornings (B;C;D;G;I) and are subsequently once again enjoying rising earlier to start their day—*“I haven’t felt groggy for a long time, you know when you get up…I hadn’t seen early morning for a long time”, “I’m able to wake up and feel more alert, like everything seems more vibrant and easy”.* One participant (G) described the better sleep-improved alertness and energy in the daytime—*“I actually thank that machine for it because it’s like my body needed it…I felt so much better for it, more alive instead of being like a walking zombie most of the day”.* Sleep hygiene and daytime napping have improved (B;E;L)—*“my sleep’s better as well, cause before it was all over the place”.* Other positive outcomes from lower levels of sleep disturbance included a perception that when they wake up, they are more productive at work (B)—*“whereas before I’d wake up and get to work and feel like I need to lie down again”.* Another participant (J) would previously get tempted to overdose on sedative medications in order to induce sleep, which is no longer happening.

*“I feel like when I’m going to sleep now, my body actually wants to go to sleep instead of like chemically having to go to sleep…normally I’d have to wait for my medication to kick in”* *(Participant D)*

*“I’m not feeling as tired when waking up. I’m still tired but it isn’t that really heavy feeling…I feel like I’ve slept”, “I know I’m gonna go to sleep rather than just lie there awake in pain…I haven’t had a nap since I started doing this [PBMT]”* *(Participant E)*

*“it makes me sleep like a baby”* *(Participant H)*

*“then I’m quite happy to just drift off again…that doesn’t normally happen, for years!”* *(Participant J)*

*“My sleep’s improved, for me, my sleep has honestly—amazing, absolutely amazing”* *(Participant L)*

*“So, before I’d go to bed and I’d be lying there until 2 o’clock in the morning…I go to bed now and I can sleep, I’m not lying there until 2 am thinking ‘what am I going to do?’…I go to bed and within half an hour I can go to sleep”, “one of the main things that affected me was sleep and that’s improved hell of a lot it really has…it’s very very rare now I wake up before the alarm”* *(Participant P)*

4: Mood lifted. A number of participants who suffered greatly from depression describe a drastic improvement in mood (A;F;K). Mood showed improvement as early as the first full treatment (K) and most significantly to the point where no longer wants to take own life (K), and new hope in a participant that had been saving to go to Dignitas prior to trial (F)—*“it gave me hope whereas before I didn’t wanna live…I do feel a bit brighter, I do”*. Participants describe feeling much happier in themselves and feeling a lot more positive in their attitude, upbeat, brighter, happier, lighter, more bubbly than normal, ‘chipper’ and *“such a boost”* (A;B;C;D; E;G;I;J;K;L;O;P)—*“my mood’s lifted and I feel good”*. One participant (A) recognises less emotional lability and subsequently being able to take on new challenges—*“I don’t you know, cry at the drop of a hat, I can take things on, my mood’s improved massively”.* Conversely, another participant (C) acknowledges that their mood is better correlated with getting tasks done. Again, participants show insight into the close linkage between symptom domains. One example is a participant (D) feeling more themselves—being able to dress how they used to because the pain is better—*“I do feel less depressed because I’m in less pain, ‘cause I’m in less pain I feel less down on myself…being able to put a bra on is obviously lifting my mood”.* A participant with post-traumatic stress disorder (PTSD) describes feeling lighter with fewer flashbacks that usually cause her to break down in tears. Participant M was struggling with anhedonia—*“I hadn’t got much interest in things, like before, if somebody gave me a newspaper/book to read I just wouldn’t bother with it”.* This disinterest has improved exponentially—*“I’d certainly say the past month I’ve started using stuff like the rubik’s cube, I mean that’s something I’ve not played on since I was a kid, so I’ve been doing something to test the mind”.*

*“I just feel as though I’m not walking through mud anymore. I feel like I can hold my head up and you know, walk on, walk straight…I don’t feel like I’ve got a cloud above my head”* *(Participant A)*

*“I mean, literally every time I come home I’ve got a smile on my face”* *(Participant G)*

*“my depression was bad, really bad, I was suicidal—but now I don’t even feel like that now…it’s improving to a state where I don’t even think about taking extra tablets or doing anything stupid…it’s definitely helped with my depression. Before I started I was suicidal, very suicidal. And it’s improved to the stage where I don’t want to commit suicide anymore”* *(Participant K)*

*“I feel a lot happy to be honest, whereas before I was I dunno just on a downer ‘cause of the pain and everything. I’m just feeling a lot happier”* *(Participant L)*

5: Reduction in analgesics and interventional therapy. Over half of participants reduced or stopped one or more medications (A;C;F;G;H;I;K;L;M;N), which included paracetamol, non-steroidal inflammatory drugs, gabapentinoids, and opioids (including tramadol, co-codamol, oral morphine solution, and morphine transdermal patch). One participant (H) stopped all analgesics completely—*“before it started I was taking so much pills, I was getting depressed about it and I just suddenly found myself missing…when I started the treatment I realised I wasn’t feeling the pain”.* Two participants (K;M) were indeed over-dosing prior to the trial intervention—*“I’ve realised I was taking too many, being depressed and in so much pain, that I was over-dosing, so I have lowered it to what it should be now and not over-dosing”.* Participants do not like using medications, with Participant I describing—*“I’ve only taken morphine twice on a night since the start of the trial so it’s much better…normally 3 times a day…I hate using it, because it makes me fall asleep and like it drains me”.* Participants identify missing analgesic doses intentionally, but in some cases unknowingly. Previously missed doses would equate to pain exacerbation, which is no longer the case as Participant L describes—*“I never took them on the weekend as I didn’t feel like I was in too much pain, whereas before when I missed it I was in a lot of pain. So, I’ve noticed without the medication, using the light therapy, the pain isn’t as much, it’s not significant whereas before it was”.* In some cases, hospital appointments were saved: Participant M’s pain was alleviated to such an extent that he went from chasing secretaries for injections with track record of minimal efficacy to now postponing injections—*“I was due appointment for facet injections, I don’t feel I need them…at this moment in time I would re-schedule, or just let me go back to my GP and let them deal with it…I don’t feel I need it”.*

6: Reduction in stiffness and improved mobility. Stiffness has generally improved (C;D;E;G;H;J;L;M;N;O), with muscles feeling much less tight and achy (D;G;J) and less spasms (E). Lower body stiffness (C), neck tension, and seizing up (E;J;M) have been abolished, joints are feeling loose again (G), shoulders, arms, and hands have increased flexibility (M;N), with a subsequent increased mobility, specifically in the mornings (C;E;L;0). Participants describe a 50–100% reduction in stiffness (G;H;J;M). One participant (J) feels as if the PBMT has penetrated into muscles and released them—*“feel softer and more pliable, seems to have really eased off”-* to an extent where she feels cured—*“sitting here now I wouldn’t think that there was anything wrong with me”.* A further participant (M) describes the neck and shoulder range of movement as the best it has even been, which has allowed him to participate in sporting activities with his wife—*“I seem to be able to move a lot better…I certainly wouldn’t be holding focus mitts for my wife you know for her to aim and punch at, so yeah I’m really happy to be fair”.*

7; Anxiety and agitation decreased. Participants are generally less anxious (A;B;D;E;G;I;L;M;O). One participant (G) with intrusive PTSD describes his anxiety level as not being very high of late. In some cases, anxiety was explained as secondary to pain being constantly at the mind’s forefront, which has consequently improved—*“I think because the pain’s not in my head so much, it’s less, I’m not as jumpy about it”.* This same participant (E) was previously heavily reliant on a medical environment, even for basic physiotherapy. This anxiety has also subsided—*“I’m not so scared I don’t think, even though I fell”.* Reduced anxiety has translated into being able to be more confident about certain activities (B;E), including those that affect livelihood—*“I’d have been anxious about driving on the motorway and stuff like that, or have to plan my route, and now I wasn’t too bad about that, so that was good”.* There is less pre-emptive anxiety, where previously a participant would brace themselves in anticipation about *“an extraordinary amount of pain”—“I’m less anxious about being in pain, because I known I’m not in as much pain I’m not having to worry…so that is making me better because I feel slightly more normal”.* Tolerance has improved in a number of participants (J;M;N;O;P), with recognition of feeling less aggravated in situations such as heavy traffic, at people asking how the pain is, and being less snappy with loved ones. In one instance (N), there has been a noticeable improvement in relationships with a grandfather being more tolerant of his grandchildren—*“I love me grandkids…it’s like I don’t want them in the house, nothing’s worse when you feel rough and somebody knocks something and you’ve gotta get up and you’re thinking ‘now look what I’ve gotta do’…but yeah, I have probably felt a little better that way”.*

8: Improved memory and concentration. Participants describe increased alertness and focus (D) and a longer attention span (B;E;G;H;I) with a new-found ability to not only engage in meaningful activities but to see them through. In one instance, Participant M describes starting the task of a rubik’s cube—something he has not done for decades—*“concentration in the past—I mean it was just like a zero, I’d just flat line on it basically. If somebody gave me something to solve I’d probably spend 2 min on it and then just say ‘no, I can’t help’…there seems to be a lot of staying power more recently”.* There is less forgetfulness in daily activities (C;E) such as misplacing keys (L), and improved word retrieval, allowing better engagement in conversations (J)—*“I don’t want to talk because I know what I want to say, it just won’t come out…I’m remembering things better…my speech seems a little more fluid”.* Participant A found working memory a real problem, to the point where she would forget where she worked or even which side of the road to drive on. She reflects that her now-improved memory has given her more confidence.

*“I’ve been reading more and not having to go back so much, ‘cause I’d have to re-read the pages all the time, so that’s been really nice…I can watch a drama and follow it through without having to rewind all the time”* *(Participant E)*

*“I’ve read half a book…I’m planning to read the other half in the next few days…I’ve been trying to read a bit more since using the [PBMT] machine because I’ve found my focus is a lot better and I can actually concentrate”* *(Participant G)*

*“Concentration’s a bit better ‘cause I’ll sit and make things. I joined an art class so I’ve been sitting and doing art so I’ve been concentrating on that…that’s been going well”, “I’ve been reading a lot…like before if I read I have to go back and read the same thing because I forgot what I already read. But I was able to read the last paragraph and sort of remember it a little bit more”* *(Participant I)*

9: Reduction in time needed to mobilise. Individuals identified a substantial decrease in time required to get ‘up and about’ especially in the mornings (B;C;E;G;I;J;L;M); partly explained by an ease in physical symptoms such as stiffness, spasm, and plantar fasciitis, but also attributed to feeling happier and more refreshed and motivated, ready to get on with the day—*“I’d sit for a good couple of hours before I got up and moved about, but now I’ll get up, have a coffee, I’ll put the washing in you know and have a potter about”* (Participant O). Participant G reported no longer having urinary incontinence, describing being able to get to the toilet much quicker—*“now, I can literally jump up and go to the toilet when I need to”.*

*“More or less 20 min, I’m up, breakfast done whereas as usually it was like an hour and a half/2 h”* *(Participant C)*

*“usually I’d lie there for 20 min before I’d even try…but I’ve been able to wake up and get straight up”* *(Participant E)*

*“I wake up…and take myself out straight away, instead of going ‘hang on, I need to stretch etc’—I can literally just jump up and do it now”* *(Participant G)*

*“When I got up this morning I didn’t think ‘oh God, the alarm’s gone off, I want to stay in bed longer’. It was like ‘oh, I’m awake now, I need to get up’.* *(Participant J)*

*“I can get out of bed…no problem at all, whereas before I’d walk as though I was crippled because the pain was just so intense”* *(Participant M)*

10: Brain fog cleared. Participant’s perceived having less ‘fog’ (A;D;E:H), less ‘groggy’ (D), ‘not as cloudy in the head’ (E), and no longer getting up in the morning feeling as though in a bubble (H). The participant now describes feeling ‘a lot clearer’ (E) with improved and clear focus (H;I). Participants describe no longer switching off and drifting off (I) and *“can actually keep my train of thought which is brilliant”* (G). Participant J identified feeling less embarrassed during conversation and finds it flows easier, whereas previously she was reduced to ‘baby words’. Participant M’s reduced fogginess has allowed both engagement with cognitive tasks and improved interactions with spouse—*“my wife’s been doing a crossword and she’s shouted across ‘have you got any input on this like?’, and I’ve thought about it, not had the answers right away, but I’ve let it tick over”.*

11: Enjoyment of body warmth. Participants described feeling comforted secondary to warmth and consequently more relaxed (D;E;J;K;M), often when symptoms can feel weather- and specifically cold-dependent (J;K;M)—*“I just feel more comfortable and warmer, more relaxed, happier...whereas with the fibro and the pain it drains you of everything…you’re sensitive to cold, achy, and miserable. I used to dread walking in the winter because the cold…if I got cold it just felt like pain”*. Muscles feel less lethargic because they are warmer (D), the body feels recuperated, and hands do not feel as bad due to warmth (D;E). Participants have felt very satisfied by these sensations and have likened them to a holiday (K;M).

*“I’ve been warm all the time which is brilliant…I love it. Everything seems easier when I’m warm”* *(Participant E)*

*“It’s like the warmth going through my body, it’s amazing, it’s like being in the sun. Lying underneath that [PBMT device] the heat starts to go through all my body because I get cold parts on my body where the pain is, which is here, here, here and here—and it heats all that up so I’m nice and cosy and warm”* *(Participant K)*

*“It’s just like a nice warm summer’s day on an Ibiza beach…I just think of good times in there [PBMT device] to be honest you know ‘cause it feels like literally in the sun, just like you’re sunbathing…it ain’t got the harshness of a sunbed, it’s just nice and warm and soothing in there”* *(Participant M)*

12: Reduced number and intensity of flares. Participants describe having far fewer bad days with regards to pain (C;D;H;P), along with reduced intensity to the point where a flare never surfaces (C) and *“not crying out in pain every 5 s”* (G): culminating in reduced prominence of pain thoughts (D) and increased engagement in activities. The new flare frequency varies between participants, reflecting the high baseline variance. In one case, ‘bad days’ reduced to one-third of previous (D)—*“these days have felt awful, but they’ve not been as awful”*, and some have had no flares whatsoever—*“I don’t have no flare up…I would normally get flare up back to back”* (H).

*“They’re like mini-flares…seems to be like what you’d feel when a flare was coming on…but then it doesn’t manifest to anything, it never gets there, it’s weird…I can’t remember when last flare was”* *(Participant C)*

*“I still have the days where…body says no, but it’s a lot fewer and far between…it ain’t too bad, they have been a bit more manageable”* *(Participant P)*

#### 3.2.2. Activities and Participation

1: Starting/re-commencing hobbies/enjoyable activities. All participants describe re-kindling previously enjoyed hobbies or even forming new ones. This is having a positive effect on mood (K). There has been a range of meaningful activities enjoyed; yoga (A), walking (A;C;E;I;J;K;L;N;O), swimming (B;M), reading (E;G), going out to theatre (C;E), arts and crafts (F;I;O) that participants describe previously losing all interest in—*“I did some stenciling with my grandchildren [for first time in 5 years]”*, baking and cake decorating (P), gardening (K)—*“I bought some hanging baskets and put them in the garden which is nice as I haven’t really bothered with it…I started to put things round there and make the garden look nice so yeah I suppose that’s turning into a hobby…that’s new”*; activities with grandchildren (F;O;P), playing with dog (G), going back to church (H), travel long distances to visit relatives (I), sparring (M), fishing (N) –*“I bought a brand new reel—she said that’s the first time I’ve seen you do anything with your fishing stuff”*, sunset horse riding trip always wanted to do (P)—*“normally when we go on holiday we hire a car and I lie by the pool…we’ve always wanted to do the horse riding…and I’ve booked it!”*. Participant E has a new-found confidence to walk her dog on her own for first time in many months—*“I’ve been able to take him out for a walk a couple of times…on my own as well…even without my stick, so that’s quite a big deal for me…I wouldn’t go on my own ‘cause I was scared of falling. I’ve done it twice this week and I plan to do it when I get home”.* Activities are looked upon with excitement and spontaneity, and are no longer eating into the next day in terms of exhaustion—*“me and my boyfriend went to the theatre on Tuesday night, I was just excited and did it. And then still got up and ready on the Wednesday—I wouldn’t have done that before. If I’d have gone out the night before I’d have just had a lazy day and done nothing…I feel like I’ve been at the theatre a lot lately, and the ballet as well”* (E). There is an increased drive to engage in hobbies –*“I felt like actually going swimming…I went twice this week for some bizarre reason”* (M). Meaningful relationships are being enjoyed again secondary to getting out with grandchildren—*“I had my grandson and we went up to Sandwell Valley…I managed to walk round the valley with him so that was good [for first time in 3 years]”* (Participant O)*, “I’ve gotta go to work and then I go home and I just wanna sit there, but we took him [grandson] to Stourport for the afternoon…I went on the little cars with him and stuff like that [shows interviewer videos] which I wouldn’t have even attempted before…I’ve got on the swings with him…I’d never do anything like that before”* (Participant P). Participant B asked a friend impromptu to join their abroad holiday, after not going away in 7 years due to fear of being uncomfortable on flight and different beds. Shopping with friends has proven to be enjoyable (D;G), with participants enjoying the present moment because they no longer have to overthink it or dread post-shopping exhaustion. Participant P cried happy tears whilst describing hosting a quiz night and food at home for whole team of colleagues and their partners—husband was in awe and extremely proud—*“well, that was me before fibromyalgia”.*

*“I’ve been able to do things I haven’t done in a long time”* *(Participant A)*

*“I’ve been twice to the pool now which I hadn’t done before”* *(Participant B)*

*“I’m taking dog out twice a day now…I can walk a bit further without my legs hurting”* *(Participant C)*

*“I’ve gone for walks just for fun with my boyfriend”* *(Participant E)*

*“I couldn’t walk long distances—I’m able to do that, and I’m not getting as tired”* *(Participant L)*

*“I’ve been swimming twice this week as well and that seems to have been easier”* *(Participant M)*

2: More able to cope with every day chores/tasks/work. There is a willingness to engage in tasks that previously would have been avoided, and there is a positive outlook on this achievement (J). Vocationally, productivity has increased (B;C;H;K)—*“I don’t have to leave work and go home early [to nap just after lunch], I can get stuff done and last until 4/5 o’clock”.* Participant H can get more done with her day due to not needing to stay in the shower for up to 2 h just to get some relief. Participant A was successful in securing a new job, and the hope that this has brought—*“I have a job to look forward to, hopefully now I can move forward and things will be OK…I think this [PBMT] has give me more of a push…I do feel more confident, confident enough to hold a job down”.* Jobs around the home are completed more easily and for longer, not only secondary to improved pain and mobility but also due to increased drive (B;C;D;E;F;G;J;K;L;M;N;P)—*“I go downstairs straight away and start doing jobs and I’m thinking ‘this isn’t like me’, it’s as if I can do things with a lot more ease”, “I’ve not used all my energy for the day doing one task”*, *“it literally feels like I’ve gone from nothing to doing everything that I can”.* Participant F is no longer needing to remain in bed for 24 h a day and has even mopped the kitchen floor for the first time in years. Participant D has been able to do simple every day things that have really uplifted her mood: she managed to wear her normal bra all week, compared with only a few hours previously. Improved grip and dexterity are described (D;I;N), including in a participant whose main hobby is arts, *“I can’t remember the last time my hands didn’t hurt long enough for me to plait my own hair…last night I was able to plait my hair”.* There is an increased drive for independence and autonomy (D;E): *“I’m less likely to rely on people. I’m trying to do things by myself”.* Participant A has felt motivated enough to start cooking homemade meals, and Participant E explains she can now drive her car due to less pain, with a clear knock-on effect on fulfillment and independence. There is less reliance on aids and adaptations in a participant that has been dependent since childhood—*“I’m hardly using it all now [stick]—which isn’t really like me. I haven’t used it [motorised bed] this week, I’ve been able to sit up myself”.* Participant C has experienced an extreme turnaround that is contributing greatly to future goals: rising earlier to walk the dog prior to work, staying in work longer hours, now volunteering to read in a school to decide whether to go back to his passion, and still having energy to walk the dog again when he gets home, as well as re-shuffle bedroom furniture and decorate in anticipation of ‘Foster 2 Adopt’, which had been put off for years due to not having energy to sort the house.

3: More willing to engage in activities with others. Participants have been able to enjoy a social life secondary to a combination of a number of improvements: consistently less pain and tiredness in the evenings for the first time in over a year (C), less social anxiety (A;E;I;L;P), increased spontaneity and ability to cope with change (N), and being able to see how much they have progressed (A)—*“I don’t get that anxiety or that you know where you just wanna scream or run and hide in a corner (laughs), I’ll just take it on board and deal with what’s coming my way”*, less exhaustion post-PBMT as compared with usual FM treatment (E)—*“my Mom’s friend came round yesterday and normally if I’d have had a treatment [medical infusion/injection] I wouldn’t want to do anything else like a few days before or a few days after”*. Participant I has been engaging in craft groups since the intervention, having the stamina to get through the session and the confidence to show her work. Participants have recognised a shift in their attitude, a desire to go out and do things that they self-confessed would have avoided previously (B;C;E;I;J;M;N), not just with those in their ‘comfort’ group, but an excitement and new-found energy to reach out to old friends (B;C;E;N)—*“3 more times yeah, and I’m going out tonight, and then next Friday we’re going out again”*, and energy and desire to help others (G), noticing they are generally more receptive in speaking to strangers in a participant with severe PTSD (G), and realising the positive knock-on effect on their mental health of being able to get out (L). Significantly, participants reported now able to complete activities in spite of the pain and have insight into this improvement (I). There is a perception of increased social stamina and engagement (A;B;C;E;G;I;J;L;N) is partly attributed to improved cognition (G;M) and has been objectively recognised by close relatives:*“[brother] ‘usually you’re like half asleep and don’t wanna talk to me, and now your rattling off and talking so much’—I’m on the phone to him for 2 h to him and then he’s like ‘I wanna go, I’m tired!’”*, with notable improved family interactions (A;B;F;G;H;J;L;M;N). Participants are confident in their bodies and that it is not going to let them down *“we’re all sharing an apartment…I’d have never done that before, before I wouldn’t share with anybody…I don’t want anybody to see me when I’m at my worst…I ain’t gonna be at my worst”,* and assertive in planning large social events—*“I said to my husband I’m having everybody from work round to the house, my team. I said I’m going to bring their husbands and we’re going to have a quiz night”.*

*“I’ve been able to socialise a bit more, so I went out for somebody’s birthday whereas before the treatment I just wouldn’t have bothered”* *(Participant B)*

*“my back was bad this week but I’ve still been wanting to do things…feeling a bit better in myself and wanting to do things, rather than not wanting to do things ‘cause I’m in pain or ‘cause I’m tired”* *(Participant I)*

*“more interactive with grandchildren…there’s been several times where I’ve took them [karate], I’ve been chatting with instructors there…me and my wife have started playing scrabble you know, that’s something again that I’ve not done in years, I’d normally have said no until these past few weeks”* *(Participant M)*

#### 3.2.3. Environment

1: Noticeable physical and emotional improvements. Participants close relatives noticed and commented on objective changes with regards to uplifted mood, noting participants are seeming more like their old selves (A;C;K;N), laughing more (A), always singing (B;K), and dancing (P)—*“I even had music on…singing…even in the office. It was actually nice to feel normal”* (in a participant that described suicidal ideation prior to intervention), more energetic (C;D;G;I) and ‘bouncy’ (C), seem happier, friendlier, ‘chirpier’ and more cheerful (B;C;D;I;K;M;P), smiling and joking much more (B). Other objective changes included a lower frequency of negative aspects of FM occurring: less ‘hunched’ and tense (D), less ‘nit-picking’ and snappy, less agitated (D;N), less ‘in on self’ (I), less jumpy in sleep—in a participant with PTSD and night terrors (D). Physical changes noted include improved walking (E;I), outwardly managing pain better (I;P), less time taken to complete activities (E), doing more including past hobbies (I;O), absence of napping (O) and improved sleep (K;P), feeling warmer to the touch (E), skin complexion better (G)—*“he said my face looked like it was glowing”*, long-term natal cleft rash has virtually vanished, confirmed by dermatologist (M)—*“normally bright red and angry and horrible”.* A range of acquaintances, including one participant’s hairdresser, describe participants as seeming ‘brighter’, in some cases as early as after the first short treatment (D;E;H;I;K). Friends and colleagues have described being more engaged in conversation (G) and more present (D). Cognitively, participants have been described as more awake (E), more alive (C), sharper and more ‘with it’ (C;J), speech seeming more fluid (J), no longer constantly asking for TV programmes to be rewound (E), and engaging in cognitive activities compared to previous anhedonia (M). There have been several instances where family, friends, support workers, and colleagues have been in awe of the changes seen; *“at work the other day I was dancing…and they was like ‘oh, what’s happened to you?!’”* (P)*, “there’s an ongoing going joke that I won’t turn up [to social events], so yeah there was a bit of a cheer when I turned up!”* (B)*, “you look 10x better, you look fantastic”* (G)*, “they [grandchildren] couldn’t believe it [when suggested stenciling]”* (F)*, “[daughter] laughs—‘you feeling alright, Mom?!’—after mopping floor”* (F)*, “[after big shopping trip] friend was like ‘you look better than I do!’, she was shocked about how much energy I had in me”* (G)*, “he was like ‘you’ve got more energy in your voice, wow you sound different’”* (G).

*“she said ‘you just seemed completely worn out constantly…it was like you were fighting, every day was a fight…but now you seem to have picked up’”* *(Participant C)*

*“it’s enabled you to do more for yourself ‘cause you’re in less pain…not dwelling on the pain…and then my Mum was saying she feels like I’ve gone from being like a victim to a survivor”* *(Participant I)*

2: Improved relationships. There has been a shift in the quality of interactions and relationships, largely secondary to the improvement seen across multiple FM symptom domains. As described in Section 3.2.1 (7), an emotional improvement has meant participants are less ‘grumpy’, angry, and on a ‘short fuse’, which has had positive effects on relationships (D;H;K;N;O). In addition to improved friendships due to now being capable of socialising, the main relationships commented on are spousal and caregiver-child. Participants were able to see the knock-on effects of one symptom domain on the next, with one (D) describing improved intimacy and bonding with partners secondary to reduced pain and pain-related agitation. One participant (M) describes a drastic increase in sexual activity with the wife—*“I mean um 4 times…yeah in 6 weeks…I mean that’s great, before that—non-existent, literally non-existent…me and my wife didn’t have a sex life…it’s been years, it’s been years and years”.* Participant P describes her husband being able to relax enough to have some beer, compared with being on ‘high alert’ due to her health. Conversations flow more easily—*“what I’ve done over the years, because I haven’t had the energy to talk, I just fluff it all by feigning listening…if I’ve had a big conversation I’m shattered”.* Carers describe being less infuriated with children, describing a consequent calmer alternate approach to discipline (H;N). Participant J describes an example involving a shift in her response to a stressful situation with her adult son, not only being able to see the funny side but also being the provider of reassurance once again. Additionally, she can find things funnier on the TV, have ‘banter’ and joke again with her son—comparing it to the previous situation where she would retreat due to being miserable in pain and feeling the need to hide it from loved ones. Children and grandchildren have appreciated the increased interaction: shopping and watching them play in the garden (L)—*“whereas before I’d say no, I’ll say yes now…before I’d stay in the car or I’d just not go out at all”*; having fun going out together again (O), opting to take grandchildren to karate and kickboxing (M)—*“well normally it’s always been my wife who takes them…I mean they normally call me ‘Grandad Grumps’ but you know I ain’t heard that phrase for quite a while…they’ve been more eager to come to me”.* Again, confidence in the body’s abilities has increased to the extent that participants can take on this role (L;M;P).

*“picked them up from school at half past 1…didn’t go to kickboxing until half 6…I was sat on sofa with my youngest grandson for 4 h…we’re sat on the sofa playing the Xbox together, I mean that’s something we’ve not done since Christmas Day, so it’s just nice building that extra bond”* *(Participant M)*

3; Insight and reduced reliance on poor habits. Participants were able to let go of learned behaviours due to feeling generally better and ‘lighter’. Cravings for sugar and alcohol were reduced (F;J), and a handful of participants subsequently reported losing weight. Participant J recognised her cravings were reduced secondary to feeling less tired and in pain: *“I think people over-eat if they are miserable in some way…because I’ve felt lighter and happier I haven’t gone into that area…I do like chocolate and I was eating a lot of it—I can’t remember the last time I had a piece of chocolate, it was probably about 2 weeks ago”, “it was my comfort, chocolate…my craving for sugar is not so bad…I haven’t been lying in bed eating chocolate”*. One participant had drunk alcohol on an evening the whole of her adult life, partly due to habit but also due to a feeling of needing to induce sleep—*“since the trial…it’s just different. Last night I thought ‘oh, I could just have one.’ and then I think ‘no, I don’t want it. I know if I have one it shall just be having it for the sake of it. My head was saying ‘well, usually at this time you have a gin and tonic’ and I thought ‘I don’t want it’.* Healthier eating habits have formed with increased motivation and energy towards home cooking (A), rising early to have breakfast with family compared with previously *“snoozing, snoozing, snoozing”.* Reducing carbohydrates and increasing vegetables and protein—this participant (H) lost 3 kg in the first few weeks of the trial. Participants have noticed more positive thought patterns.

#### 3.2.4. Whole-Body Photobiomodulation Therapy

1; Positive PBMT experience. A range of experiences of the intervention were described: brilliant (E), enjoyable (B;L;N), very relaxing (A;I;J;K;L;P), soothing (D), comfortable (D;I), warmth (I;J;K), and participants felt excited to come to each session (A;E;I;K;L;M;O)—*“I’m itching to go for the next session”*, compared with the usual feelings of not looking forward to the experience of an injection (A;B). Participants enjoyed both ‘just lying there and letting their mind drift off’ (A;J;L) as well as how they feel afterwards (D;E;G;H;J;K;L)—*“it’s like a little treat on the NHS”, “it’s like a sunbed”.* No side effects were described.

*“It’s not only the benefits you get from the thing, it’s like you’re switching off from the world for 20 min, and I think it’s the routine that helps as well because you don’t do that at home or work”* *(Participant B)*

*“I wish I could take that machine home with me and use it every day [laughs]”* *(Participant G)*

*“I look forward to it…I didn’t wanna come off today, yeah I do get excited about coming…the 20 min was absolutely amazing, it was like being in the sun and really helped with my moods…like I say, it got me through the day at work”* *(Participant K)*

2; Attributable changes to PBMT. Participants attributed a range of positive outcomes directly to the intervention, describing a relief that something is actually working (A;B;C;E;L;M)—*“I’m sure it’s to do with that [points to device] because it wasn’t like that before”.* Participants feel shocked at some of the effects they have experienced. One example of this is less pain (G;H;L;M)—*“when I use it, it literally feels like somebody’s melted away all the pain”, “it’s beyond my expectation”, “since this [PBMT] I can’t believe there’s just no pain there you know and I just don’t know what it’s done, I mean I just don’t know what…I’ve come off the injections, I mean if somebody had said ‘that will clear’ I’d have said ‘get away’…I’d had it so long and it had got to the point where I was having injections every 3 or 4 months, the doctor was saying we can’t keep giving injections…it will cause problems with your tendons, so it weren’t a solution”.* Further positive results were directly attributed to PBMT are: improved sleep (K;L;P)—*“I’m just having the best sleep—honestly I dunno [laughs] what it is about this treatment”*, heightened energy levels (J), a positive shift in mindset and ‘lightening mood’ (G;J;K;M;P)—*“I’m surprised it’s lifted my mood completely”, “always cheers me up…it always makes me feel a bit brighter”, “my depression’s improved with that [points to device] ‘cause I don’t know what it is about light, it lifts me up, it makes you feel happier”*. In some cases, this effect was extreme—*“the light therapy, it definitely has probably saved my life…I mean without it I wouldn’t be here now”*. One participant (J) describes a definite positive change even after the first 6-minute treatment but struggles to put it into words:*“on the evening I felt different, there’s not a word for it, I was very aware of it as well, but I couldn’t pinpoint anything…I still can’t describe it, it’s something I’ve never felt before”.*

*“Everything just feels better and easier—they’re my over-riding comments…it’s made such a difference”* *(Participant E)*

*“It’s like my brain, a switch has gone, something’s happened to it, I don’t know how that light has done it—it’s done something…I haven’t felt like that for a long time, years, and I’m wondering if it’s that machine”* *(Participant J)*

*“I think this machine has done the majority of the work because I would never have had the range of movement in the upper body…there’s something going on because I wouldn’t have had this change in my sex life you know…my sister-in-law had said I’m seeming much chirpier, she wouldn’t have said that…but this light therapy just seems to have—now I’m able to get out of bed in the morning, walk to the toilet and have no pain in my feet, I mean I can get up out of this chair”* *(Participant M)*

3; Recommendation to others. Participants would recommend PBMT to others: A;B;D;E;G;H;J;K;L;M;N;O;P)—“without a shadow of a doubt” (D), “I hope other people have the chance to do what I’ve done and I hope it helps other people like it’s helping me” (A). Some see PBMT as being far superior to conventional treatments—“it’s worth a try because all the painkillers in the world haven’t done for me as much as light therapy’s done for me” (G). Participants have described PBMT as a “lifesaver” and a participant (F) who had been saving money to go to Dignitas describes “it gave me hope, whereas before I didn’t wanna live…it’s give me hope that there could be something that could come from this that can help me have a better quality of life and improve my health and wellbeing”. Participant M describes “jumping at it” if given the chance for a further 18 treatments.

4; Fear of treatment ending. This sub-theme describes fear of treatment ending secondary to positive outcomes experienced, even to the point of no longer feeling suicidal, which was quickly attributed to PBMT (K). Participants describe feeling sad (B) towards the end of the trial, going to miss the treatment, feeling at a loss, fear of “going downhill”, and needing to “go back on those pills” with no future access to PBMT (A;B;D;H;N)—*“it’s become my little routine now and I don’t know how I’m going to cope once this has stopped”*. Participant D felt aggrieved at this prospect—*“part of me doesn’t wanna see too much of a benefit, for it then to have to just stop, I dunno it’s like giving a child a toy and then taking it back off them”*, and Participant K was very worried given the perceived drastic improvement in mental health seen. One participant (N) felt that there should be a plan for a few “top up” treatments on the NHS—*“is there any plan where you can come in and have a boost?...just probably have 6 and then that would keep you going for another 5 or 6 months probably”.*

*“I’m under no illusion now that I haven’t got the bed to look forward to I’ll be putting the analgesic doses back up again”* *(Participant A)*

*“I’ve had a little bit of relief but how long will it last?”* *(Participant F)*

*“I mean I’m actually going to cry today because this is the last one—I wanna take it home with me”* *(Participant G)*

*“What happens at the end of the trial—can we carry on with it?”* *(Participant I)*

5; Unanticipated effects. Participants experienced a range of positive outcomes that were not anticipated by themselves or the investigators. This ranged from 4 kg of weight loss (H), improvements in skin conditions and complexion (L;M)—*“I’ve had effects, they’ve been like other effects which have been good…like my skin seems to have improved…I’ve got no dry skin...and I had like dark spots for years which have faded as well… honest to God, it’s crazy!”, “bags under eyes as well…they used to be really dark black to the point where you’d think somebody had punched me in the eyes…those have got better now as well—something’s going on and I just can’t explain it”,* Plantar fasciitis 100% gone by week 4 (M)—*“suffered with it for years and years and years…and that seems to have magically almost gone…it’s almost non-existent, I’ve suffered with it quite bad, tried lots of different injections and things”*, toes are no longer numb (O).

### 3.3. The Thematic Synthesis

#### 3.3.1. The Bridge to Recomposition

During the analysis stages, after delving deeper beyond the initial sub-themes in line with OMERACT domains [38], four further and novel sub-themes were developed that were seen to positively bridge the initial sub-themes into the recomposition phase (Figure 2). Again, these are presented in order of commonality.

1; Increased motivation. All participants had insight into their improved motivation and energy for life, with a change in mindset and a willingness to perform activities—*“the energy has given me the motivation to do stuff”,* with participants directly describing and recognising their own increased motivation. Participants attribute their improved symptoms to their new-found motivation (E;L)—*“cause my body feels different, you know with the warmth—so it makes me wanna try…’cause my pain’s less I wanna try things more”,”before I just couldn’t be bothered to do certain stuff…now, I’m just doing it without thinking”*; translating into increased activities (A;B;E;F;G;I;J;K;L;M;N;O;P)—*“I’m going to go fishing…it will be the first time I’ve gone in probably 3 years”*, *“I seem to be cooking more homemade meals…I do seem to want to do that more”, “well my wife goes kick-boxing twice a week, what if I hold the focus mitts for her?”.* One participant describes no longer wanting to nap due to fear of missing out on things, while her motivation and energy are higher. Prior to the trial intervention, Participant K had felt suicidal and had no drive to perform basic personal hygiene tasks—neglecting for several consecutive days. Now, the same participant describes, *“I’m washing, showering every day, washed me hair last night, teeth are being brushed every day…done the house from top to bottom—I don’t do that! I did the windows, I did vacuuming, I stripped the beds and didn’t need to stop for a rest”.* Participant M has gone from seeing a professional for cognitive issues relating to long COVID, observing high levels of disinterest, to now *“playing some mental games, just to see if it could improve my cognitive and memory recall—it’s just something I felt like doing…I’ve been YouTubing looking how to solve it and stuff like that”.* There was a reported increased drive towards wanting to talk to people and visit friends, where this would have previously been avoided. Looking towards the long-term, participants have felt a “kick-start” from the PBMT, putting their change in mindset into action towards life-changing events; wanting to go back to work, doing voluntary work with a view to getting back into teaching passion; de-cluttering the house; getting ready for fostering; finally finishing a degree that has been put off for 10 years—*“I found myself yearning to go back and study and finish my coursework”* (C;D;H). Again, Participant F has seen a drastic change from lying in bed and saving for Dignitas to an indescribable feeling of the desire to get out and do things—*“I’m looking around and thinking I need to do something, I don’t know what but I just want to do something, such as the plant pots. Sitting there doing that was a big achievement for me, and mopping the floor. But I feel now I want to do a little bit more, I’m trying to make an effort”.*

2; Feeling proud/sense of achievement. Participants felt proud for a variety of reasons: from obtaining a new job (A), going walking (E;I;J), gardening, including carrying a half-full watering can (I), to calling their best friend to share that they have been able to wear a normal bra for 3 days (D), with a range of achievements in between (G;J;K;M). This sense of fulfilment instilled hope and confidence to aspire to continue to achieve going forwards—*“I felt really proud of myself I did, really really proud of myself and it made me realise I’ve been missing out on so much of my life”.* Participant P felt exhilarated at hosting the work’s quiz night and became emotional when describing her husband’s reaction:*“he was just sat in the corner like that and he was just watching me and he was like…he was really proud”.*

3; Increased confidence. There was a new-found confidence in the physical body, leading to a conviction and assuredness that tasks would be completed (D;E;I;J)—*“I’ve been able to take dog out for a walk…on my own as well, so that’s quite a big deal for me, it had been years…I wouldn’t go out on my own ‘cause I was so scared of falling…I feel more confident when when I’m walking without my stick, that’s a massive thing for me, they gave me 2 sticks when I was 16. I’m 39 now”*. Participants would previously “second guess” their ability to engage in activities for fear of pain, tiredness, and the time taken to recover. Participant D feels “more free to go and do something” and even “more present”. In one instance, this involved multi-tasking under challenging circumstances, while simultaneously enjoying it. Participants now feel they have the capability to try new tasks (M). Furthermore, confidence has increased secondary to improved cognition and adaptability in conversations (A;J)—this has culminated in Participant A going as far as to get a new job during the intervention, now realising confidence has further increased due to getting back to some sort of normality—*“I’ve got a job you know and that’s boosted my confidence as well…now confident enough to hold down a job…I’m just a different person, I’m walking out of here today a much more confident person than when I first came in”.* A shift in mindset towards getting back to previous hobbies has meant improved relationships (N;P)—*“my daughter likes her fishing as well and she’s mentioned about coming”, “I felt confident enough to be there for him [grandson]”.*

4; Feeling more like ‘old self’. Participants describe relief as feeling more like themselves for the first time in years (B;E;G;J)—*“I haven’t felt like this for 3 or 4 years, and you’d pay anything just to feel like that again”*, having previously felt like they had lost themselves due to the pain, constantly making excuses and telling themselves they could not do things (G)—*“I don’t feel like a disabled person as much in that sense”*. For some, there were practical elements such as getting back to prior-loved hobbies (G;I;P)—*“but like now I feel like I used to…that was me before the fibromyalgia [organising quizzes, big social events]…it was just normal for us”*, and the simplest daily achievements proved to be extremely empowering, bringing hope (D)—*“it’s quite uplifting [being able to wear normal bra compared with sports bra]…these little wins are making me feel more normal*…*simple things like curling my own hair”,* and being able to laugh like they used to (J). Family and partners have been relieved to see these changes (D;J;N;P)—*“happy because he can see there’s something I used to do is coming back”.*

#### 3.3.2. Describing the Upward Recomposition Spiral

Themes and sub-themes are reported initially as fairly separate entities. In reality, they heavily rely on one another, and these processes are starting to be realised and discovered by the participants themselves. We describe a life-changing recomposition phenomenon (Figure 3) where new processes are being shown to amalgamate into a meaningful output that has left the participant with a new ability to begin to lead a new type of life, with coping strategies to keep moving forward in spite of knockbacks, compared with their prior low threshold to cycle back round into the downward decomposition spiral.

The following and final sub-themes are the pinnacle of the recomposition process, highlighting a complete shift in mindset, outlook on life, and coping strategies—again, presented in order of commonality.

1; Ability to cope with and push aside other symptoms. Participants are beginning to feel very positive about life and believe in their ability to cope with it despite the unpredictability of FM symptoms (A;B;D;G;K;M;N)—*“I’m actually getting through the day without having to phone my boss and say ‘look, I need to go home’”.* There has been less catastrophising about symptoms, with more rational thinking on what symptoms mean for daily living, and what is more is that participants themselves are recognising this and, in a sense, coaching themselves out of it (C;D;I)—*“I can get on with what I want to, so if I have bad days it’s ‘well, actually they’re not gonna last forever’. Whereas before it was like ‘oh God, is it coming, how long’s this gonna last for’”, “it’s not a forward thought anymore, it’s just ‘OK, you’re in pain but you can still keep doing things’”.* One participant (F) with a particular negative outlook previously now describes doing more with her day thanks to her change in mindset—*“I can’t spend the rest of my life lying in this bed, thinking, and lying there in pain. But you’re gonna be in pain whether I’m sitting in the garden or lying down”.*

2; Self-awareness and insight into symptom interlinkage. Participants describe a cloud being lifted (B) and pain being less intrusive and easier to deal with, in part due to improvement in other symptoms, for example, heightened energy, mood, and sleep (A;B;D;I;J;K;L;M)—*“muscles softer and more pliable…I thinks that’s overall why I’ve been feeling so light and happier”*. Participants are recognising the positive cycle of pain alleviation, allowing them to be active and get out and see people, which in turn helps pain and mental health (C;E;N). Participant E’s mood was elevated secondary to the psychology of no longer relying on a stick.

3; Improved pain and sleep having substantial effect on mood, energy, confidence, motivation and ability to cope. This sub-theme represents a positive feedback loop where improved coping, confidence, and motivation in turn strengthen the already improved pain and sleep (B;E;I;J;L;M;N;P)—*“I just felt happier and wasn’t in so much pain so I thought I might as well try while I can and then if it doesn’t work, it doesn’t work, but at least I tried”.* Participants describe a willingness to engage in and adapt for tasks—while finding a sense of achievement in doing so (J). There is confidence, resilience, and motivation to plough through a task despite failure and with new-found stamina (M). These effects have translated into increased work productivity (A;B;C;K;L), and cleaning the house for a whole day. The major take-home message for this sub-theme is that a real problem-solving attitude and stamina have developed.

4; Improved sleep and, more importantly, improved sleep hygiene. There have been improved sleep habits, possibly as a consequence of engaging oneself in daytime activities and feeling appropriately tired at night (D;F;G;I;J;O). There has been a drive to stay awake during the daytime, resulting in healthier sleeping patterns (E;G;I)—*“but now I’ve been going to bed like half 9/9 o’clock, and sleeping through until 7 o’clock. I have been staying awake so that when I go to bed I am sleeping…I haven’t had a nap since I started doing this [PBMT]”.* There has been a change in mindset and less reliance on the habit of taking morphine or alcohol in an attempt to induce sleep or scrolling through the phone for several hours in bed (A;I;J;P)

5; New-found enjoyment in hobbies secondary to improved memory and concentration. Participants have taken pleasure in being able to resume reading a book or watching a TV drama (C;E;G;I). In one case (Participant M), there has been a ‘U-turn’ from not being bothered to pick up a book or newspaper, to presently enjoying challenging himself with brain-training games—*“it still ain’t solved yet, I’ve done the 1 face and the botton 2 rows but I’m just like scratching my head over it, but you know before I wouldn’t even have tried”*. For Participant A, low confidence secondary to poor memory is no longer a major obstacle to social integration. Participant I’s carer describes the pain not being at the forefront of the mind, so creativity has been able to come through once more, and the participant seems more herself.

## 4. Discussion

We believe our study is the first in the field of chronic pain management to utilise qualitative methodology to directly assess the acceptability and efficacy of a specific medical intervention in a clinical trial. The objective of this work was to investigate whether qualitative data had value in describing the experience and effects of a course of whole-body PBMT in an FM cohort. The data obtained is abundant and rich, which indeed confirms the value of this methodology in assessing the utility of a treatment, particularly one that is so novel. This work builds on our previously identified ‘Fibromyalgia Decomposition Phenomenon’ [29], where we liken FM processes following diagnosis to resemble a complex and heterogeneous ‘breakdown of a compound into multiple products’ [39]. That is, the ‘breakdown of a person as a whole into smaller segments that no longer appear to work together in unison’ [29]. We now unveil the potential within these participants to not only return to good functioning but, in many instances, to go beyond this and engage with life in an enthusiastic and meaningful way. The precipitator that has tipped this balance positively into the ‘Fibromyalgia Recomposition Phenomenon’ is seemingly a course of whole-body PBMT. It would not have been possible to deduce these two novel phenomena with quantitative data alone.

### 4.1. The Fibromyalgia Recomposition Phenomenon

This work has undergone thematic analysis to bring out a multitude of themes and sub-themes. We then delved much deeper to identify more complex and interlinking processes that culminated in the ‘Fibromyalgia Recomposition Phenomenon’. As previously described, prior to trial intervention, participants exhibited a potential to enter the recomposition cycle, albeit small, with most data representing individuals’ despair at the hand they had been dealt. We describe recomposition in this context as the reorganisation of individuals’ constituents, being remodelled or recycled into new ways of being [29], following the process of decomposition. All of our participants are seen to readily enter the recomposition cycle described in Figure 3. Every facet and symptom domain have improved. Most importantly, outlook on life has seen a dramatic shift—*“tomorrow, regardless of how I feel, I have got things to do…you just gotta get on with it…having this condition is something you’ve just gotta get on and live with”.* Some negative cases were noted during the initial sub-theme analysis, namely, ongoing aches and pains, residual stiffness, some sleep disturbances, occasionally still feeling fatigued, and mental struggles due to ‘family crises’. Interestingly, no negative cases were identified in the ‘processes’ that entailed hobbies, coping, engaging with others, improved motivation and confidence, and feeling proud and like their ‘old self’. These processes are the culmination of initial sub-themes and are arguably the most significant factors in regaining considerable physical and psychological functioning. FM possesses a 65% lifetime prevalence of depression [40], yet several studies proceed to preclude trial entry if the subject suffers from major depressive disorder [41], potentially introducing bias. A particular positive finding from our study was the ‘real-world’ approach to recruitment—89.5% of participants commenced the trial with severe FM and 26.3% with ‘severe’ depression. In spite of severity at baseline, all participants demonstrated entry into and through the recomposition phase.

### 4.2. Proposed Mechanisms of Whole-Body PBMT in the FM Cohort

We recently stipulated proposed mechanisms of action of whole-body PBMT in FM: reduction in oxidative stress in a condition known to encompass increased reactive oxygen species secondary to mitochondrial dysfunction; improved oxygen delivery to muscles that are known to have reduced and altered capillary supply; and dealing directly with widespread small-fibre neuropathy associated with FM pain and sensitivity [22]. Of course, this is in addition to the well-known PBMT effects of reducing systemic inflammation and analgesic mechanisms, including a reduction in myofascial pain [42,43,44]. Secondary to our qualitative data, we are able to further speculate on proposed mechanisms of action in this cohort. Firstly, the dramatic effect on psychological functioning we have witnessed in our population is mirrored in multiple studies. There is an abundance of growing evidence of PBMT showing benefit in a multitude of central nervous system conditions and states, including but not limited to: anxiety and depression [45,46,47], reduced drug cravings [47,48], bipolar disorder [49], traumatic brain injury [50], age-related cognitive decline [51,52,53,54,55,56,57], including working memory [58] and executive function [59], dementia [56,60], Parkinson’s disease [61], and improved neuroplasticity in ischaemic stroke recovery [62]. Our data has highlighted the strong interlinkage between mental state and physical functioning. Additionally, PBMT is known to have direct effects on improved sleep [63]. Therefore, the positive outcomes seen are partly secondary to the complex ‘diamond’ of improved mental health having widespread knock-on effects on daily functioning and integration back into life, which then feeds back into improved mental health by way of a positive feedback loop. 

It is difficult to describe any one factor at play to account for this seemingly exponential positive recomposition cycle. There are many subleties interacting. For example, PBMT has been shown to produce direct increases in endogenous endorphins [64]. Much like the aetiology of FM, endorphins act on both the peripheral and central nervous systems, peripherally exerting opioid-like effects and inhibiting pain transmission and centrally causing increased production of dopamine while activating pain modulatory descending pathways [65]. Glial cells are being increasingly recognised as producing and maintaining chronic pain states [66]. Both endorphins [64] and brain-derived neurotrophic factor (BDNF) [67] are shown to stimulate hippocampal neurogenesis. *“PBM therapy is capable of coaxing stressed neurons into producing neurotrophic factors BDNF and glial cell line-derived neurotrophic factor (GDNF), (p. 958)”* [68]. The anterior cingulate cortex (ACC) has been identified as an important relay in the medial pain pathway, which, importantly, integrates the affective and motivational components of pain [69]. It is known that opioid drugs have action at the ACC, which can result in not necessarily lower pain scores but see the patient caring less about their pain [70]. This is one potential scientific explanation as to why a small cohort of our participants seemingly have similar post-treatment pain scores, yet their functioning is dramatically higher. PBMT studies support the notion of ACC involvement [71]. 

The above factors demonstrate just a small proportion of PBMT’s potential centrally acting mechanisms and warrant further investigation in relation to FM in their own right. Secondary to a combination of factors, our participants have been enabled to engage in increased levels of physical activity, which in turn further boosts endorphin levels [72]**,** leading to positive mental health and attitude [73]. This improved mental health and physical activity have a subsequent positive effect on relationships. Endorphins, amongst other hormones and neurotransmitters, have been referred to as “social neuropeptides” and show increases secondary to improved relationships and can influence social behaviour [74]. This propels participants at an acceleratory pace through this recomposition phase. Each interlinkage becomes stronger further on in the process.

### 4.3. Limitations

One participant missed the mid-treatment interview; however, data from the mid- and post-intervention interviews were grouped. We could not control for ‘real-world’ factors; two participants self-professed to wanting to keep the final interview short due to feeling unwell and recent bereavement, respectively. Several treatments were postponed in a patient who developed COVID, and another with a bout of allergic rhinitis. With this being a feasibility trial with a small sample size, there was no placebo control. Therefore, we cannot rule out the placebo effect of a participant simply being on the trial and getting back into a routine. In that regard, it is possible that the interviews themselves have acted as a therapeutic modality, which has been recognised in previous research [75]. The same authors recently analysed baseline interviews for the same participants [29]. Secondary to this prior experience, it is possible that our vision was already shaped, and we could see processes coming out earlier and clearer. As aforementioned in previous publications [22,29], pain catastrophizing was not formally assessed. The interview process itself, reflecting the added time spent with participants compared with a routine clinical consultation, was seen to unveil insight into, for example, symptom interlinkage—*“because it’s gradual, I don’t see it. And then I realise when I speak to you”, “it’s intriguing because it’s nice to have a doctor ask these questions”, “somebody’s sat down with me and talk to me about things which I’ve never had that before and I’ve found that helpful”.*

### 4.4. Recommendations for Future Research and Clinical Implication

Going forward, it is vital that qualitative methodologies to assess responses to pain interventions are routinely utilised. Our work alone shows that quantitative and qualitative data bring out different results within the same setting, on the same day, with the same investigator. Within this small sample alone, two participants quantitatively scored their pain as not much improved, which did not correlate to their much-improved quality of life and function when assessed qualitatively. The Pain Catastrophizing Scale (PCS) will be a particularly useful tool towards measuring this exaggerated negative orientation towards pain [76]. These particular participants would have likely scored highly on the PCS, and therefore its use would aid in the quantification of outcome measures. Rating scales possess inherent limitations [77,78] and individuals can become overwhelmed and overloaded with multiple questionnaires, especially when multimodal in nature. The trust and rapport gleaned during interviews reflect a more real-world experience. Furthermore, we should be adopting this proven therapeutic interview style approach during consultations. It is otherwise impossible to delineate the myriad of biomedial, psychological, and behavioural factors in order to shape and individualise treatment plans [79]. Qualitative tools are deemed a more compassionate approach [25], at a time when compassion is something FM patients feel can be lacking in healthcare settings [80]. Future studies should consider qualitative data collection beyond 3 months to satisfy NICE recommendations for research [27]. A 15% worldwide FM prevalence [81] is significant, and with the Royal College of Physicians’ recent push towards higher rates of community diagnosis [1], this is only forecast to have an upward trajectory. It is therefore more important than ever before to have effective and easily accessible therapies. A recent large (*n* = 941) study addressing FM patients’ perspectives revealed fewer side effects and higher acceptability for non-pharmacological treatments compared with pharmacological treatments [82].

## 5. Conclusions

This study demonstrates the richness that qualitative data brings when evaluating a treatment. The authors’ conclusion is that the strength and consistency of these results among participants simply cannot be ignored, and we owe it to FM patients to pursue further definitive research on this safe and non-invasive treatment modality with a view to it eventually becoming a commonplace therapy widely available to those who need it. Until now, no pain treatment has demonstrated such a pronounced shift in all facets of not just livelihood but also increased stamina and resilience and new ways of approaching life. All participants, prior to the present trial, demonstrated evidence of ‘The Fibromyalgia Decomposition Phenomenon’ [29]. This was in spite of having tried a multitude of both NHS and non-NHS treatments. We now show that PBMT possesses efficacy in a multitude of pain and psychological conditions—FM is a condition that encompasses a wide spectrum of these, and therefore lends itself to whole-body PBMT. This is corroborated by our data, which evidently shows participants positive and acceleratory entry up the recomposition spiral. Participants’ significant and real fear of their whole-body PBMT coming to an end nicely sums up the urgent need for this treatment to be made available in the NHS.

## Figures and Tables

**Figure 1 biomedicines-12-01116-f001:**
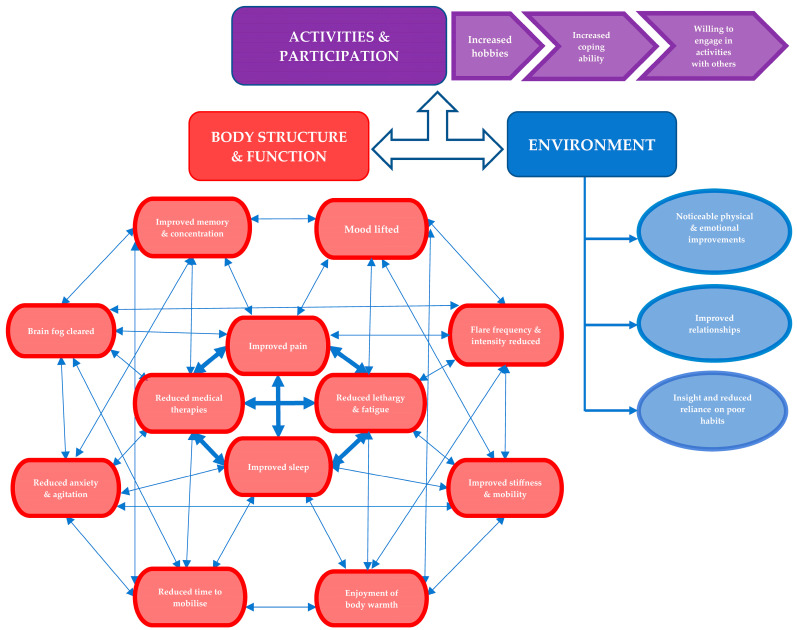
Themes, with 18 sub-themes, derived from the study’s dataset. The ICF model is again depicted, as described in the baseline interviews [29]. Here, the ICF domain ‘Body Structure & Function’ can be likened to the OMERACT FM Working Group hierarchy [38]: the inner circle is akin to OMERACT ‘core domains’; the outermost circle represents the outer ‘skin’ of the OMERACT onion. The ‘Body Structure & Function’ domain can be seen as a ‘diamond’ appearance, reflecting the positivity now seen as compared with the prior ‘web’. Note, two of the ‘core domains’—‘Patient Global’ and ‘Multidimensional Function’ are represented later in the process description.

**Figure 2 biomedicines-12-01116-f002:**
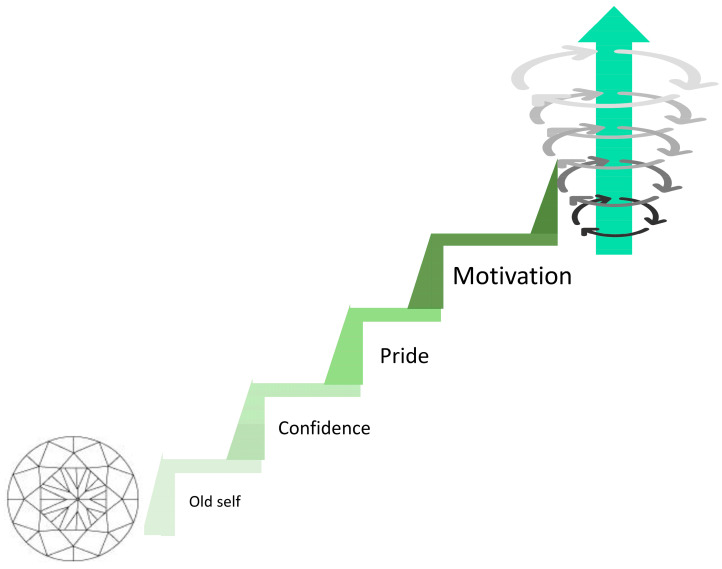
Upwards step bridge of the culmination of initial sub-themes giving rise to positive characteristics and behaviours. The diamond (bottom left) represents the initial sub-themes described in Figure 1, which have leant themselves to overall changes in character traits during the trial—shown as the step bridge towards the recomposition cycle (top right). The traits are presented in order of commonality, with increased motivation being most commonly observed.

**Figure 3 biomedicines-12-01116-f003:**
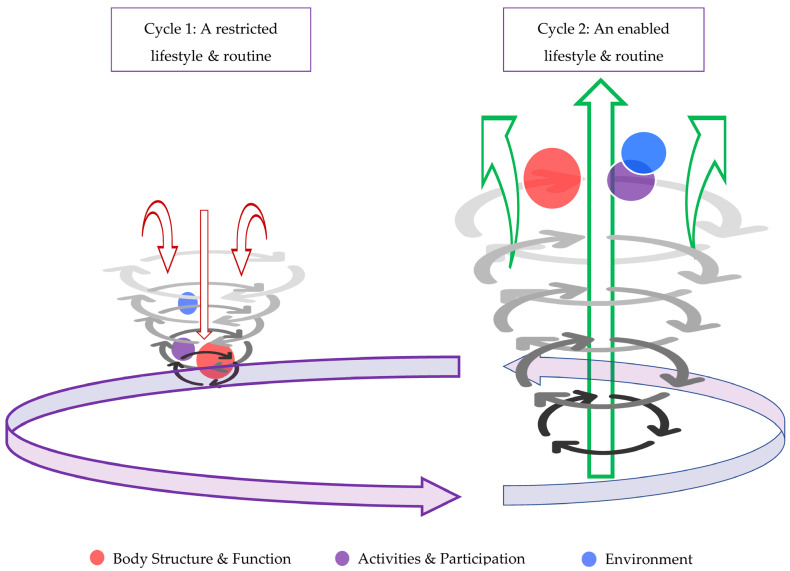
The recomposition phase. In contrast with the decomposition phenomenon witnessed at baseline, a positive cycle is now emerging and features prominently in participants’ everyday lives. In fact, the balance has tipped drastically towards recomposition, according to interview data, with a 6.3%:93.7% decomposition:recomposition ratio in comparison to the 98%:2% seen at baseline. The spirals are different sizes to more easily illustrate this. The ICF domains (represented by the coloured circles) are sized comparably smaller, with the baseline representing less intrusiveness. They can actually be seen almost emerging from the spiral altogether. Furthermore, the horizontally cycling purple arrows have favourably switched, with a much lower propensity for an individual to return to the negative decomposition spiral, especially after a ‘knock-back’.

**Table 1 biomedicines-12-01116-t001:** Eligibility criteria.

Inclusion Criteria	Widespread chronic pain of any origin (including axial pain, polyarthralgia, and myofascial pain)
	Able to provide informed written consent
	≥18 years
	Able to commit time to the trial treatment schedule of 6 weeks
	Score as low or moderate risk on the COVID-19 risk stratification tool—applicable for the duration of the pandemic
Exclusion Criteria	Pregnancy
	Score as high risk on the COVID-19 risk stratification tool—applicable for the duration of the pandemic
	Body weight ≥136 kg
	Uncontrolled co-morbidities (e.g., uncontrolled diabetes defined as HbA1c >69 mmol/mol, decompensated heart failure, major psychiatric disturbance such as acute psychosis or suicidal ideation)
	Use of systemic corticosteroid therapy including oral prednisolone or corticosteroid injections within the preceding 6 months
	Known active malignancy
	Inability to enter the NovoTHOR^®^ device or lie flat for 20 min (either due to physical reasons or other e.g., claustrophobia);
	Individuals speaking a language for which an interpreter cannot be sought (Oromo, Tigranian, Amharic, Greek)

**Table 2 biomedicines-12-01116-t002:** Template for intervention description and replication (TIDieR) checklist.

Brief Name	➢ Whole-Body Photobiomodulation Therapy—18 sessions.
Why	➢ Eighteen sessions are the currently recommended, widely instituted, accepted practice with the NovoTHOR^®^ device. ➢ This device was developed in 2013, and since then, 406 NovoTHOR^®^ systems have been developed, of which 300 remain in regular use, treating at least four patients per device per day. This equates to approximately 2.6 million treatments since its inception. No significant adverse events have been reported to date.
What	➢ All participants entering the trial received a course of whole-body PBMT. ➢ The NovoTHOR^®^ Whole-Body PBMT device consists of a hinged, clamshell design with light-emitting diodes (LEDs) arranged to emit near-infrared and visible red light → PBMT is delivered to the entire body at once. ➢ A Participant Information Sheet (PIS) was provided at least 48 h before participants consented to the study. They were given the opportunity to undertake an experience session. ➢ Participants were expected to lie horizontally in the device with the lid as closed as they were comfortable with.
Who provided	➢ All trial investigators, following a short training session on the use of NovoTHOR^®^. Audio-recorded semi-structured interviews were conducted by the investigators.
How	➢ The LED equipment delivers red and near-infrared light therapy to the participant. All interviews took place face-to-face in the same clinical setting as the intervention.
Where	➢ Clinical Research Facility, SWB NHS Trust. Participants are registered at the trust and are therefore geographically within the region. ➢ The device requires a well-ventilated, spacious, temperature-controlled room, with appropriate mains electricity. Interviews took place pre-, mid-, and post-intervention over the course of 6 weeks.
When and how much	➢ Session 1 = 6 min. Session 2 = 12 min. Sessions 3–18 = 20 min. ➢ Timescale: 3 treatments/week for 6 weeks. ➢ The dosage of LED light (also known as ‘fluence’) will be equivalent to 25 J/cm^2^. The device will supply a dual wavelength of red and near-infrared light with a 50:50 ratio of 660 nm and 850 nm, respectively, via 2400 LEDs.
Tailoring	➢ After liaison with experienced clinicians within the field with experience dealing with our population in the NovoTHOR^®^, we decided to slowly uptitrate the treatment times during the first three treatments for all participants.
Modifications	➢ There were no modifications to treatment. Some participants expressed a wish for a handrail to help get themselves in and out.
How well	➢ All participants received 18 treatments; however, some participants took longer than 6 weeks to complete the course due to unforeseen circumstances.

**Table 3 biomedicines-12-01116-t003:** Demographic data and characteristics of study participants.

Demographics and Characteristics	n (%)	Mean ± SD	Median (IQR)
Sex			
Female	14 (70)		
Male	6 (30)		
Age (years)		47.3 ± 10.9	49 (41–53)
Symptom duration (years)		15.6 ± 7.7	14.5 (10–20)
Marital status			
Married	10 (50)		
Single	6 (30)		
Divorced	1 (5)		
Co-habiting	2 (10)		
Civil partnership	1 (5)		
Employment status			
Employed full-time	4 (20)		
Employed part-time	1 (5)		
Self-employed	2 (10)		
Unemployed (looking for work)	1 (5)		
Unemployed (not looking for work)	7 (35)		
Sick leave	1 (5)		
Retired	4 (20)		
Education level			
Some secondary school	1 (5)		
Completed secondary school	2 (10)		
Completed further education (sixth form)	1 (5)		
Higher education	16 (80)		
Ethnicity			
Asian or Asian British	5 (25)		
Black British	1 (5)		
White British	14 (70)		

## Data Availability

The datasets used and/or analysed during the current study are available from the corresponding author upon reasonable request. The data are not publicly available due to Confidentiality issues.

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
