# Peer review of "Whole-Body Photobiomodulation Therapy Propels the Fibromyalgia Patient into the Recomposition Phase: A Reflexive Thematic Analysis"

_biomedicines, 2024, doi:10.3390/biomedicines12051116_

Round 1

Reviewer 1 Report

Comments and Suggestions for Authors

The manuscript entitled “Whole-Body Photobiomodulation Therapy Propels the Fibromyalgia Patient into The Recomposition Phase: A Reflexive Thematic Analysis”. An interpretive hermeneutic phenomenological study situated within the worldview of pragmatism was undertaken. 

Below are some suggestions:

In the Abstract:

- The abstract should be rewritten with a clearer introduction and objective. The authors need to better describe the methodology as well as the results obtained;

- My suggestion regarding the abstract is to insert a paragraph demonstrating the relevance of the manuscript, according to the objective presented;

- Insert a clearer conclusion in line with the objective.

 1. In the Introduction:

I suggest the authors include in the last paragraph the justification for carrying out the research, as well as the objective, which was not clear.

2. Materials and Methods:

2.1. Qualitative approach and research paradigma: The authors should adjust this item, the sentences are redundant and unclear as to the scientific contexto;

2.4. Sampling strategy: Insert a flowchart indicating the number of participants, inclusion and exclusion criteria....is easier for readers to see;

2.10. Techniques to enhance trustworthiness: The title of the table should go before the table (as a header) and be better described;

3.3.2. Describing the upward recomposition spiral: Adjust the legend in figure 3 (overlay);

3. Results

3.1. Description of participants: The title of table 2 should go before the table (as a header) and be better described; 

3.2. Themes: Adjust the colors and size of the letters in Figure 1, as it is difficult to see. 

4. Discussion 

- The authors should begin the discussion by contextualizing the research, as well as its objective and main results. 

5. Conclusion 

The conclusion should begin with a summary of what has been achieved for the reader's positioning, as well as clearer data in line with the research objective. It's too redundant.

*******The references do not comply with the journal's standards.

Comments on the Quality of English Language

Moderate editing.

Author Response

Thank you for your important comments and suggestions which have improved the manuscript. Below a summary of your comments are given together with author responses in italics.

In the Abstract:

- The abstract should be rewritten with a clearer introduction and objective. The authors need to better describe the methodology as well as the results obtained;

- My suggestion regarding the abstract is to insert a paragraph demonstrating the relevance of the manuscript, according to the objective presented;

- Insert a clearer conclusion in line with the objective.

       - Thank you for your suggestions. An objective has been added into introduction. The methodology and results have been expanded. Finally, the relevance of manuscript has been added and a new conclusion provided to reflect objective.

  1. In the Introduction:

I suggest the authors include in the last paragraph the justification for carrying out the research, as well as the objective, which was not clear.

       - Thank you for identifying this. Justification for the work is provided and followed by an aim and objective clarified in last paragraph

  1. Materials and Methods:

2.1. Qualitative approach and research paradigma: The authors should adjust this item, the sentences are redundant and unclear as to the scientific context;

       - The sentence has been removed thanks.

2.4. Sampling strategy: Insert a flowchart indicating the number of participants, inclusion and exclusion criteria....is easier for readers to see;

       - A table has been inserted. Subsequent table numbers have been altered.

2.10. Techniques to enhance trustworthiness: The title of the table should go before the table (as a header) and be better described;

       - Table titles have been moved to before tables and expanded on.

3.3.2. Describing the upward recomposition spiral: Adjust the legend in figure 3 (overlay);

       - Adjusted – thank you

  1. Results

3.1. Description of participants: The title of table 2 should go before the table (as a header) and be better described; 

       - Moved and expanded on.

3.2. Themes: Adjust the colors and size of the letters in Figure 1, as it is difficult to see. 

       - Figure sized increased, font altered, italics removed. Thanks.

  1. Discussion 

- The authors should begin the discussion by contextualizing the research, as well as its objective and main results. 

       - 2 sentences inserted at beginning of discussion addressing research context and objective.

  1. Conclusion 

The conclusion should begin with a summary of what has been achieved for the reader's positioning, as well as clearer data in line with the research objective. It's too redundant.

     - Summary inserted and authors position explained

Reviewer 2 Report

Comments and Suggestions for Authors

The work is of high clinical interest, well presented and described.

I suggest improving the abstract, as it is unclear in the methods and results part.

It would be useful to explicit in the manuscript (test or tables) the questions that composed the interview. Were they chosen basing on previoulsy published/validated questionnaires? If not so, please explain why.

Author Response

Thank you for your comments and identification of how the manuscript can be improved.

I suggest improving the abstract, as it is unclear in the methods and results part.

  • Thank you for this comment. An objective added into introduction. Methodology and results expanded. Explained further the relevance of manuscript. A new conclusion to reflect the objective has been written.

It would be useful to explicit in the manuscript (test or tables) the questions that composed the interview. Were they chosen basing on previoulsy published/validated questionnaires? If not so, please explain why.

  • Thank you. This has been explained in section 2.6. data collection methods

Round 2

Reviewer 1 Report

Comments and Suggestions for Authors

I thank the authors of the manuscript for making all suggestions during the review process.

Comments on the Quality of English Language

Moderate editing.